# Preparation and Optical Study of 1-Formamido-5-Isocyanonaphthalene, the Hydrolysis Product of the Potent Antifungal 1,5-Diisocyanonaphthalene

**DOI:** 10.3390/ijms24097780

**Published:** 2023-04-24

**Authors:** Erika Kopcsik, Zoltán Mucsi, Bence Kontra, László Vanyorek, Csaba Váradi, Béla Viskolcz, Miklós Nagy

**Affiliations:** 1Institute of Chemistry, University of Miskolc, Miskolc-Egyetemváros, 3515 Miskolc, Hungary; ria.toth1@gmail.com (E.K.); kemvanyi@uni-miskolc.hu (L.V.); bela.viskolcz@uni-miskolc.hu (B.V.); 2Advanced Materials and Intelligent Technologies Higher Education and Industrial Cooperation Centre, University of Miskolc, Miskolc-Egyetemváros, 3515 Miskolc, Hungary; kemcsv@uni-miskolc.hu; 3Department of Chemistry, Brain Vision Center, Liliom utca 43-45, 1094 Budapest, Hungary; bence.kontra@femtonics.eu; 4Department of Organic Chemistry, Semmelweis University, Hőgyes Endre utca 7, 1092 Budapest, Hungary

**Keywords:** naphthalene, fluorescence, solvatochromic effect, isonitrile, formamide, DFT

## Abstract

Aromatic isocyanides have gained a lot of attention lately as promising antifungal and anticancer drugs, as well as high-performance fluorescent analytical probes for the detection of toxic metals, such as mercury, even in vivo. Since this topic is relatively new and aromatic isocyanides possess unique photophysical properties, the understanding of structure–behavior relationships and the preparation of novel potentially biologically active derivatives are of paramount importance. Here, we report the photophysical characterization of 1,5-diisocyanonaphthalene (DIN) backed by quantum chemical calculations. It was discovered that DIN undergoes hydrolysis in certain solvents in the presence of oxonium ions. By the careful control of the reaction conditions for the first time, the nonsymmetric product 1-formamido-5-isocyanonaphthalene (ICNF) could be prepared. Contrary to expectations, the monoformamido derivative showed a significant solvatochromic behavior with a ~50 nm range from hexane to water. This behavior was explained by the enhanced H-bond-forming ability of the formamide group. The significance of the hydrolysis reaction is that the isocyano group is converted to formamide in living organisms. Therefore, ICNF could be a potential drug (for example, antifungal) and the reaction can be used as a model for the preparation of other nonsymmetric formamido–isocyanoarenes. In contrast to its relative 1-amino-5-iscyanonaphthalene (ICAN), ICNF is highly fluorescent in water, enabling the development of a fluorescent turnoff probe.

## 1. Introduction

Isocyanide, or the isonitrile group (–N^+^≡C^−^ ↔ –N=C:), is the real chameleon of organic chemistry [1]. It is found in both natural compounds and synthetic drugs [2], it can serve as a ligand in metal complexes [3], and may be a versatile precursor in isocyanide-based multicomponent reactions [4], such as the Ugi reaction. Depending on reaction conditions and its partner, it can react as a normal triple-bonded –N^+^≡C^−^ functional or as a carbene radical –N=C: [5]. This unique behavior makes isonitrile one of the most useful building blocks of “smart materials”, such as photochromic complexes [6], supramolecular networks [7], and environmentally sensitive dyes [8]. Moreover, isocyano groups are isoelectronic to cyano groups and could substitute for them in specialty applications, such as liquid crystals [9,10].

Utilizing the strong electron withdrawing (EW) nature of the isonitrile group, Nagy et al. recently developed a novel class of push-pull fluorophores, where the isocyanide group is connected to an aromatic amine [11,12,13,14,15,16,17,18,19,20,21,22]. The resulting intramolecular charge transfer (ICT) gives rise to excellent solvatochromic properties, as we previously demonstrated through 1-amino-5-isocyanonaphthalene (ICAN) [11]. The spectrophysical properties are easily controlled by the relative positions of the –NH_2_ and –NC groups [12], as well as the substituents (reactive [14,16] or inert [13,18,22]) on the –NH_2_ group or by exchanging the aromatic core [19,20,21]. The synthesis of ICAN derivatives is cheap and straightforward since the respective aromatic diamine is reacted smoothly with in situ-generated dichlorocarbene (:CCl_2_) in basic solution [11]. This reaction yields both the nonsymmetric amino-isocyano (ICAN) derivative and the symmetric diisocyano (DIN) derivative as well. However, using dichlorocarbene, the yield of the diisocyano derivative is usually very low; therefore, an alternative synthetic route such as the formylation and subsequent dehydration of the starting diaminoarene molecule is needed when the diisocyanoarene derivative is the product of interest. Until now, our focus was on the nonsymmetric products (ICANs) because of their excellent solvatochromic properties.

However, recently, we demonstrated that 1,5-diisocyanonaphthalene (DIN) could be very effective in the treatment of multidrug-resistant *Aspergillus* and *Candida* strains [18,22]. Since *Aspergillus fumigatus* is one of the most lethal fungal infections, especially for COVID-19 patients [23], the understanding of the solution state properties of DIN is of paramount importance. This knowledge may encourage the development of novel isocyanide-based antifungals that could be welcome ahead of the next viral pandemic. Despite the large number of bioactive isocyanides of natural origin [24,25,26], the isocyanide moiety has so far received little attention from a medicinal chemistry standpoint, mainly due to the tenet that isocyanides are chemically and metabolically unstable. Galli et al. recently demonstrated [27] that primary aliphatic and aromatic isocyanides undergo enzymatic hydrolysis, yielding the respective formamide derivative. This conversion of the isocyanide group can also happen upon the action of acids in aqueous media [5,28] and may be utilized in the more acidic interior of lysosomes in cancer cells. We have shown that our isocyanoacridine derivatives are promising photodynamic anticancer agents [20], which accumulate in the lysosomes. The isocyanide–formamide conversion may play a significant role in the photodynamic effect. The investigation of this conversion and the properties of the resulting formamides are therefore of paramount importance.

Although the hydrolysis reaction is well-known, much to our surprise no one has tried to utilize it until now for the preparation of aromatic formamido-isocyanides, the synthesis of which is very challenging. Interestingly, even the very simple molecule 1-formamido-5-isocyanonaphthalene (ICNF, Figure 1) has not been prepared yet, according to our literature search (SciFinder structure search, March 2023).

Here, we report the first and facile synthesis of N-(5-isocyanonaphthalen-1-yl)formamide (ICNF) as the hydrolysis product of the medically important 1,5-diisocyanonaphthalene (DIN). The optical study of both DIN and ICNF were carried out with particular focus on special dye–solvent interactions. The experimental findings were supported by high-level quantum chemical calculations. We believe that our results may contribute to the better understanding of the behavior of isocyanides in various environments, such as the inside of living cells. In addition, as Bruffaerts et al. recently pointed out [29], ICNF can serve as a valuable formamide precursor for the selective preparation of a broad range of urea-type compounds, carbamates, and heterocycles via ruthenium-based pincer-complex catalyzed acceptorless dehydrogenative coupling reaction.

## 2. Results and Discussion

### 2.1. Synthesis of 1,5-Diisocyanonaphthalene

The most efficient method for synthesizing symmetric isocyanides such as 1,5-diisocyanonaphthalene (DIN) is the formylation of the primary amine (1,5-diaminonaphthalene, DAN), followed by dehydration of the resultant formamide with phosphorus oxychloride (POCl_3_) in combination with bases, most commonly tertiary amines. The dehydration reaction gives high yields in a number of solvents, and Samali et al. recently reported a novel cosolvent-free method, where near 100% isocyanide yields could be obtained in pure triethylamine (TEA) [30]. It should be noted, however, that a number of papers list dichloromethane (CH_2_Cl_2_) as an effective solvent for the dehydration reaction. In the case of DIN, it proved to be completely unusable, and we had to use tetrahydrofuran (THF) instead. After consulting with other synthetic groups, we realized it was a known issue, but unfortunately no one had pointed it out until now.

N,N′-(naphthalene-1,5-diyl)diformamide (NDF) was obtained from naphthalene-1,5-diamine by the reaction of the amino groups with formic acid, then it was converted to 1,5-diisocyanonaphtalene (DIN) by dehydration of the formyl groups using phosphorous-oxychloride. In order to study the reaction of DIN with acids, N-(5-isocyanonaphthalen-1-yl)formamide (ICNF) was synthetized by carefully reacting DIN with trifluoroacetic-acid. The synthesis of the dyes in this study is presented in Figure 1. The purity and structure of the compounds were checked by ^1^H and ^13^C-NMR. ESI-MS measurements further confirmed the structures as the measured m/z values differed by no more than ±0.01 Da from those of the ones calculated (NMR and HRMS spectra can be found in the Appendix A).

### 2.2. Optical Characterization of 1,5-Diisocyanonaphthalene (DIN) and Related Computational Studies

To study the ground and excited state electronic properties of the 1,5-diisocyanonaphthalene (DIN) via UV–vis, excitation and emission spectra were recorded in a variety of solvents selected to cover a broad range of solvent polarity values, spanning from the non-polar hexane to the polar H_2_O. The solvents were selected to cover the possible specific dye–solvent interactions, such as H-bonding, too.

The UV–vis and fluorescence spectra of DIN recorded in acetonitrile are presented in Figure 2, while the excitation and steady state emission spectra in various solvents are presented in Figure 3. The absorption maximum wavelength (λ_Abs_) and the corresponding molar extinction coefficients (ε) as well as the emission maxima (λ_Em_) are compiled in Table 1. All the UV–vis spectra in various solvents are presented in the Appendix A.

As it is evident from Figure 2 and Figure 3A, the UV–vis and the excitation spectra are identical for most solvents, indicating the absence of specific solvent–dye interactions (H-bonding or ESIPT) in the excited state. A broad absorption band can be identified between approximately 250 and 330 nm (40,000–30,300 cm^−1^). The structure of this absorption band is virtually the same as that of naphthalene (Figure 2, inset), however found at lower energies. It has a symmetrical structure and consists of five peaks. The most intense one is located at 300 nm (33,333 cm^−1^), while the absorption maximum of naphthalene is located at 275 nm (40,000 cm^−1^), indicating a possible electronic interaction (partial delocalization) between the aromatic electrons of the naphthalene core and the triple bonds of the isocyano groups. The bathochromic shift is also present in the case of the peaks next to the maximum (266 to 288 nm and 286 to 314 nm in the case of the left and right peaks, respectively). The molar extinction coefficients (at 300 nm) vary between 11,000 and 15,000 M^−1^cm^−1^ (Table 1), which is in very good agreement with the theoretical calculations (Figure 2 inset). These values are more than double compared to those of naphthalene (6000 M^−1^cm^−1^ at 275.0 nm) [31], further supporting the presence of a larger delocalized electron structure, 10 e^-^ in naphthalene versus 14 e^-^ in DIN. DIN exhibits good fluorescence in most of the solvents (Φ_f_ = 13–21%). It should be noted here that quantum yield could not be calculated in water due to the appearance of a novel band at λ_em,max_ = 423 nm. This band belongs to the hydrolysis product and will be discussed later.

Interestingly, fluorescence was completely quenched in DMSO ((Φ_f_ = 0.5%), and very low fluorescence was observed in pyridine (Φ_f_ = 0.7%) and DMF (Φ_f_ = 1.1%), too (Figure 3A,B). This phenomenon is strange since all our previously prepared isocyanonaphthalene (ICAN) derivatives were highly fluorescent in both DMSO and DMF [11,12]. Another strange behavior was observed in toluene, where instead of the double peak emission a broad charge transfer character band with a maximum at λ_em,max_ = 352 nm (Figure 3B) appeared. Since no visible change in the UV–vis spectra in the case of DMSO, DMF, pyridine, and toluene could be detected (Appendix A) compared to the other solvents, the explanation for the quenching and bathochromic shift may be electron transfer between the dye and the solvent in the excited state. This phenomenon has yet to be understood and investigated in detail, and therefore will be the topic of a follow-up article.

The position of both the absorption and emission peaks remained virtually the same in every solvent (Table 1, Figure 3A,B), since the dipole moment of DIN was 0 in both the S_0_ and S_1_ states owing to the symmetrical build of the molecule. The non-solvatochromic behavior and relatively high energy emission bands of DIN are advantageous for practical applications. When one or two of the isocyano groups are transformed in a reaction, the emission is expected to redshift due to the loss of the strong electron-withdrawing NC groups. Symmetry break of the molecule is more favorable since the resulting amino or formamido groups (from NC) are electron-donating, resulting in the formation of push–pull-type solvatochromic dyes such as 1-amino-5-isocyanonaphthalene (ICAN) or 1-formamido-5-isocyanonaphthalene (ICAF), as will be detailed in the following sections of the paper.

To understand the theoretical details of the photophysical processes related to DIN as well as ICNF (see later), we computed the ground state (S_0_), the vertical excitation (S_1_*), and the relaxed extricated state (S_1_). The DIN molecule represents a symmetrical structure, which is retained after the vertical excitation (S_0_ → S_1_*). The change in the transition electric dipole moment (Dip.S.) is around 3.0 Au. The major transition is purely related to the excitation from HOMO (46th) to LUMO (47th) orbitals. Dip.S. and all the other values, such as wavelength (λ) and energy gap (Δ*E*), depend on the relative permittivity of the solvent applied (Figure 4). The change in transition electric dipole moments is attributed to the simultaneous geometrical rearrangement of the two -NC group during the relaxation at S_1_ state, where the C–N bond is strengthening (*d*_1_, shortening bond distance by ca. 0.025 Å), while the N≡C bond is somewhat weakening (d_2_; elongating bond distance by ca. 0.003 Å) as shown in Figure 5. In fact, the ground state naphthalene ring represents a distorted and a less aromatic structure. During the excitation, the C–C bond distance in the naphthalene ring balances out to the aromatic average (*ca* 1.42 Å). This rearomatization process is supported by the slight increase in the aromatic character of the nucleus-independent chemical shifts (NICS) [32] of the two rings in S_0_ →S _1_ (−10.4 ppm → −10.9 ppm). The aromaticity percentage is also slightly increased in water from 182 to 186% relative to benzene during vertical excitation [33]. The method was applied analogously before in the case of excitation of chromophores [34,35].

The Milliken group charges of the isocyanide groups were partially negative (−0.079) in the ground state, whose average structure can be represented by the two left-hand-side resonance structures. Here, the molecule was polarized by the two negative functionalities [2 × (−0.079)], and an electron deficiency in the aromatic ring (Σ = +0.158). In the excited state (S_1_), the weight of the two right-hand-side structures was elevated, which means a 2 × −0.033 electron density transfer into the aromatic ring, resulting in a less polarized structure.

### 2.3. The Hydrolysis of 1,5-Diisocyanonaphthalene and the Formation of 1-Formamido-5-Isocyanonaphthalene (ICAF)

When investigating the emission spectrum of DIN in water, we noticed the appearance of a novel band centered around 415 nm. This band was completely absent in the spectra recorded in other solvents. It is known in the literature [27] that the isocyano group is converted to formamide as a result of hydrolysis or enzymatic action in living organisms. Therefore, we assumed the 415 nm band belonged to a formamide derivative. To test our assumption, a small amount of 10% (m/m) HNO_3_ was added to the aqueous solution of DIN and spectra were recorded every 1.5 min. The results are presented in Figure 6A. After the addition of the acid, a sudden drop in the fluorescence intensity of DIN at 330 nm can be observed; however, the intensity of the 415 nm band only slowly increased and reached saturation after 20 min, while the signal of DIN constantly decreased during the 75 min measurement time. An isosbestic point could be detected at 380 nm, indicating the presence of two fluorescent species. The hydrolysis reaction was surprisingly slow considering that the solution contained approximately 0.005 M HNO_3_ (pH~2.3).

The synthesis of the monoformamide derivative is quite challenging since until now no one has described it in the literature. (SciFinder search for 1-amino-5-formamido-naphthalene resulted in 0 results in March 2023) Therefore, we tried to optimize the reaction conditions so that one of the isocyano groups could be selectively converted to formamide and to obtain the monoformamide derivative in pure form.

Into the acetonitrile solution of DIN (1.197 × 10^−5^ M), a small amount of trifluoroacetic acid (TFA, 0.10 M) was added and the changes in the solution were monitored by fluorometry using a 314 nm excitation wavelength.

It is evident from Figure 6B that the signals of DIN decreased continuously during the reaction and in the initial phase a new signal appeared near 400 nm, which increased with time. An isosbestic point can be observed, which is characteristic of the presence of two fluorescent species. However, after a certain time, both the DIN and the new signals started to decrease (Figure 6B inset). This phenomenon can be explained by the initial formation of the fluorescent monoformamide derivative ICNF, when only one isocyano group reacted with the oxonium ions in the solution (Figure 7). During the second part of the reaction, the remaining isocyano group was also converted to formamide, resulting in the formation of the nonfluorescent diformamide NDF (Figure 7 and Figure 8A).

### 2.4. Optophysical Properties of 1-Formamido-5-Isocyanonaphthalene (ICNF) and the Related Computational Study

UV–vis spectra of ICNF recorded in different solvents reveal that the exchange of one isocyano group of DIN to formamide resulted in significant changes in the absorption properties. The results are summarized in Figure 9A and in Table 2. Instead of the rigid, 5 peak absorption band of DIN, independent of solvent polarity, a broad absorption band is seen in the range of ~270–350 nm (or ~37,000–28,500 cm^−1^, Figure 9A), which can be attributed to the intramolecular charge–transfer transition (ICT) between the donor formamido and the acceptor isocyano groups. 

It is clearly seen from the data of Table 2 that λ_Abs_ is dependent upon the solvent polarity; i.e., the ICT absorption bands suffer a slight redshift with increasing solvent polarity. The bathochromic shifts (from n-hexane to DMSO) are approximately 10 nm (1120 cm^−1^) for ICNF, while only 3 nm or 330 cm^−1^ was observed in the case of DIN. The shifts of the low energy bands are indicative of the polar character of the ground state. Indeed, DFT calculations yielded 5.67 and 6.89 D ground-state dipole moments in hexane and DMSO, respectively. It should be noted, however, that the symmetric product 1,5-diisocyanonaphthalene does not have a dipole moment in either ground or excited state, which explains the lack of ICT band and the structured naphthalene-like absorption spectrum. It can also be surmised that besides dipole moments, specific dye–solvent interactions, such as H-bonds, may also be responsible for the shape and position of λ_Abs_ [36]. The presence of specific interactions becomes more obvious when we compare the shape and maxima of the absorption and excitation spectra presented in Figure 9A,B. Without solvent, the absorption and excitation spectra should be identical. In the case of ICNF in H-bond donor solvents such as dioxane, THF and DMSO the absorption spectra contain a shoulder at higher wavelengths (320–330 nm) besides the maximum located at approximately 310 nm (Table 2). In the excitation spectra, the maxima are shifted 8–10 nm bathochromically (Figure 9B, dashed lines) as these shoulders become the maxima, for example, in the case of THF λ_Abs_ = 314 nm and λ_Ex_ = 314 nm. In contrast, in H-bond donor solvents (dotted lines in Figure 9B) such as isopropanol and methanol, λ_Abs_ and λ_Ex_ values are virtually the same and are found below 310 nm. These values are lower than would be expected based on the dielectric constants of the solvents and even lower than measured in the case of hexane, the least polar solvent in Table 2. The effect is more striking in the case of water (strongest H-bond donor), where both λ_Abs_ and λ_Ex_ are blueshifted to 295–296 nm, which is 15 nm lower than the value of hexane. It can be concluded, therefore, that the optical properties of ICNF are strongly affected by H-bond formation both in the ground (S_0_) and in the excited (S_1_) states. The H-bond is formed between the formamide moiety and the solvent, and it is energetically more favorable when ICNF is the H-donor (redshift), compared to the case when it is the acceptor (blueshift).

In contrast to DIN, ICNF showed a significant solvatochromic behavior. The steady-state fluorescence properties are presented in Figure 9C,D and are summarized in Table 2. The solvatochromic range was approximately 50 nm (λ_Em, hexane_ = 375 nm−λ_Em, water_ = 418 nm). The Stokes shifts varied between 5600 cm^−1^ in hexane and 9800 cm^−1^ in water. The compound was highly fluorescent in most solvents; quantum yields (Φ_F_) were 40–70%. Interestingly, in hexane the Φ_F_ value dropped to 11% and ICNF fluorescence was almost completely quenched in pyridine (Φ_F_ = 0.5%). The reasons may be the dimer formation in hexane and H-bonded complex formation in pyridine, as will be discussed later. We previously showed the strong H-bond-forming capability of amino-isocyanoarenes with pyridine [13]. Of course, the shifts are additionally influenced by the solvent polarity due to dielectric stabilization.

The ground state (S_0_), the vertical excitation (S_1_*), and the relaxed excited state (S_1_) were also modeled (Figure 5) analogously to DIN. The 1,5-ICNF structure is not symmetrical and not planar, due to the amide functionality, which turns out of the plane of the naphthalene ring. The vertical excitation (S_0_ → S_1_*) represents also undoubtedly a pure HOMO (51)–LUMO (52) transition, with a moderate transition electric dipole moment. The absorbed and emitted wavelengths (λ) and related energy gaps (Δ*E*) varied by the relative permittivity of the solvent applied. After the geometrical rearrangement at S_1_ state, both the C–N and N≡C bonds were shortening remarkably by 0.013 Å and 0.021 Å, respectively. Moreover, the ground state twisted amide bond turned back almost completely to the plane of the naphthalene ring (34° → 6°). The Milliken group charges of the isocyanide groups was also partially negative (−0.089) at the ground state, like for DIN. In the excited state (S_1_), the CN group transferred electron density into the aromatic ring, represented by the slightly increased NICS value in the right-hand-side ring of the naphthalene (−10.4 ppm → −10.8 ppm).

One of the most common ways to quantify the solvatochromic effect in solvents of different polarity is to plot the fluorescence emission maxima (ν_Em_) as a function of the empirical solvent polarity parameter E_T_(30) [37].

Interestingly, two groups can be identified in Figure 10a: one belonging to the protic (H-bond donor, blue) solvents with OH groups (iPrOH, MeOH and H_2_O) and the other to the mostly H-bond acceptor, yellow (THF, dioxane, pyridine, DMF, DMSO) solvents. In both cases, the correlation is linear between the fluorescence emission maximum and E_T_(30). It can be surmised from the corresponding slopes that ICNF exhibits significantly stronger dye–solvent interactions with H-bond-acceptor solvents (148 kcal^−1^cm^−1^mol) than with H-bond-donor ones (85 kcal^−1^cm^−1^mol).

In order to quantify the solvent effect on the solvatochromic parameters such as emission maxima, and Stokes shifts, multiple linear regression (MLR) analysis employing the Catalan scale [38,39] was used according to the equation:(1)$$y=y_{0}+a_{\mathrm{SA}}\mathrm{SA}+b_{\mathrm{SB}}\mathrm{SB}+c_{\mathrm{SP}}\mathrm{SP}+d_{\mathrm{SdP}}\mathrm{SdP}$$
where y_0_ is the property of the substance of interest (e.g., emission maximum and Stokes shift) in the absence of solvent, for example, in the gas phase. SA is the quantitative empirical measure of the ability of bulk solvent to act as a hydrogen-bond donor towards a solute. SB is the quantitative empirical measure of the ability of a bulk solvent to act as a hydrogen-bond acceptor or electron-pair donor towards a solute, forming a solute-to-solvent hydrogen bond or a solvent-to-solute coordinative bond, respectively. SP and SdP are the solvent polarizability and dipolarity parameters, respectively, determined using reference dye molecules. a, b, c and d are the corresponding coefficients and their inclusion in the equation indicates the dependence of the property under investigation upon the respective solvent parameter. The values of solvatochromic parameters were collected from [14]. The results are presented in Figure 10b and in Table 3.

As seen in Figure 10b, the Catalán equation describes perfectly the specific interactions, and based on the data of Table 3, solvent polarity has the largest effect on the solvatochromic behavior but both the H-bond-donating and -accepting capabilities of ICAF are also pronounced (large a and b parameters) because of the presence of the formamido group.

### 2.5. Analysis of ICNF Fluorescence Quenching by Pyridine

It is assumed from the solvatochromic investigation of ICNF that it exhibits specific dye–solvent interactions, presumably H-bonding in solutions. To better understand this phenomenon, pyridine titrations in three different solvents (Hexane, MeCN and THF) of largely different H-acceptor properties were carried out. Pyridine was chosen since it is known that fluorophores with a NH moiety tend to form HB complexes with pyridine. The results of the fluorescence titrations are presented in Figure 11A–C. It is evident from the spectra that pyridine effectively quenched the fluorescence of ICNF in all three solvents in the order of THF > MeCN > Hexane. The addition of 10 µL pyridine (0.04 M) reduced the fluorescence intensity by 25% in THF, 43% in MeCN, and 65% in hexane. However, the fluorescence of ICNF was almost completely quenched in pure pyridine; only 1% of the FL intensity measured in MeCN remained. The maximum wavelengths of fluorescence emission of ICNF showed no change in THF and MeCN; however, a slight redshift (from 375 to 383 nm) was observed in hexane.

To describe the fluorescence quenching (dynamic or possible H-bonding interactions), we used the linear Stern–Volmer equation (Equation (2)).
(2)$$\frac{F_{0}}{F}=1+K_{\mathrm{SV}}\left[Q\right]$$
where F_0_ and F are the intrinsic fluorescence intensities of ICNF in the absence or presence of the quencher (pyridine), respectively. [Q] is the concentration of the quencher, and K_SV_ is the Stern–Volmer quenching constant. Figure 11A–C insets show the respective Stern–Volmer plots. In the case of THF and MeCN, the plots are almost perfectly linear up to 1 M (approx. 7% *v*/*v*) pyridine content. Linear Stern–Volmer plots indicate that a single class of fluorophores exists in the solution, and that only one mechanism (dynamic or static) of quenching occurs. 

Surprisingly, in hexane the SV-plot is linear only up to 0.1 M pyridine content and above that it shows a very distinct downward curvature, which could indicate the presence of two populations of dye residues where one of them is not accessible to the quencher. The inaccessible or buried residues are responsible for the remaining fluorescence. In the case of the monoformamide ICNF, good solubility is not probable due to the presence of the polar and strongly H-bond-forming formamide groups. We assume that ICNF forms some kind of H-bonded associates in hexane. This could also explain the reduced fluorescence intensity compared to other solvents. Only half the fluorescence intensity was obtained in hexane compared to that in THF or MeCN. The other sign may be the two distinct shoulders on the emission band at around 360 and 400 nm. In other solvents, the emission band was symmetrical (Figure 9C).

The Stern–Volmer constants calculated from the slope of the plots were 7.89 ± 0.2 in THF, 18.76 ± 0.2 in MeCN, and 38.51 ± 0.2 in hexane. The large difference in the K_SV_ values may be a proof of H-bond formation between the dye and the solvent. Where the H-bond is strong, for example in THF, pyridine must compete with the solvent. On the other hand, in hexane the free dye population is easily accessible for pyridine, explaining the four times higher K_SV_ value.

According to the theoretical calculations, the dimerization in hexane as solvent is quite an exothermic process from an enthalpy point of view (Δ*H* = −53 kJ mol^−1^; Figure 12) and this benefit remained in the case of Δ*G* as well (−4.9 kJ mol^−1^). The calculated absorbance and emitted wavelength values (Table 2) differed little from those of the monomer form, which are in agreement with the experimental observations.

Pyridine bonds also formed a remarkably strong HB with the amide NH, resulting in exothermic Δ*H* (−36.7 kJ mol^−1^) and Δ*G* values (−5.7 kJ mol^−1^; Figure 12). This supports the obtained titration curve. The quenching mechanism was a proton transfer to the pyridine ring.

In addition, theoretical calculations could explain the unusual splitting of the signals in the ^1^H NMR spectrum of ICNF (Appendix A). Since the rotation around the C-N bond in amides is usually hindered, the appearance of two isomers (rotamers), namely cis and trans, is expected (Figure 1 and Figure 12). Based on DFT studies, the energy difference between the two forms is 0.6 kJ·mol^−1^, the cis rotamer having lower energy. The 0.6 kJ·mol^−1^ difference agrees well with the experimental cis:trans ratio of 70:30 determined from the signal ratios in the ^1^H NMR spectrum of ICNF.

## 3. Materials and Methods

Acetone, dichloromethane (DCM), hexane, 2-propanol (iPrOH), toluene, ethyl-acetate (EtOAc) (reagent grade, Molar Chemicals, Hungary) were purified by distillation. Acetonitrile (MeCN), tetrahydrofuran (THF), methanol (MeOH), dimethyl formamide (DMF), dimethyl sulfoxide (DMSO), pyridine (HPLC grade, VWR, Germany), cyclohexane, 1,4-dioxane (reagent grade, Reanal, Hungary), were used without further purification. Other solvents and reagents were purchased from Sigma Aldrich in reagent grade and used as received. Deuterated solvents were purchased from Eurisotop.

The reactions were monitored by a Shimadzu LC-40D XR UPLC-MS system equipped with a SIL-40C XR autosampler, SPD-M40 photo diode array detector, and an LCMS-2020 DUIS mass spectrometer operated in negative and positive ionization modes. The separation was carried out on an Ascentis^®^ Express C18, 2 μm UHPLC column (L × I.D. 5 × 2.1 mm), at 40 °C provided by a CTO-40s column oven. Gradient elution was used with either eluent 0.1% TFA in water (A) and 0.1% TFA in MeCN (B); or 0.4 g NH_4_HCO_3_ in 1 L water (A) and MeCN (B).

For preparative HPLC, an Armen SPOT Prep II instrument using UV detector (200–600 nm scan) equipped with a Phenomenex Gemini C18, 250 × 50.00 mm; 10 µm, 110A column was used. Gradient elution was employed using 0.4 g NH4HCO3 in 1 L water (A) and acetonitrile (B) as eluent system.

NMR: NMR spectra were recorded at 25 °C in the solvent indicated, either on a Varian Mercury Plus spectrometer (Agilent Technologies, Santa Clara, CA, USA) at a frequency of 400 MHz (^1^H) or 101 MHz (^13^C), or on a Varian Unity INOVA spectrometer operating at a frequency of 500 MHz (^1^H) or 126 MHz (^13^C). Notations for the ^1^H NMR spectral splitting patterns include singlet (s), doublet (d), triplet (t), broad (br), and multiple/overlapping peaks (m). Chemical shifts of the resonances are given as δ values in ppm and coupling constants (J) are expressed in hertz.

UV–vis: The UV–vis spectra were recorded on a UV-6300PC double beam spectrophotometer (VWR International) in a quartz cuvette of 1.00 cm optical length. A 3.00 cm^3^ solution was prepared from the sample. Background was recorded for the pure solvent in the reference cuvette.

HRMS: HRMS spectra were recorded on a Xevo G2-XS QTof mass spectrometer, with the default ESI ionization method. 

Fluorescence Measurements: Steady-state fluorescence measurements were carried out using a Jasco FP-8550 fluorescence spectrophotometer equipped with a Xe lamp light source. The excitation and emission spectra were recorded at 20 °C using 2.5 nm excitation, 2.5 nm emission bandwidth, and 200 nm/min scanning speed. Fluorescence quantum yields (Φ_F_) were calculated by using quinine-sulfate as the reference, using the following equation:(3)$$\Phi_{F}=\Phi r*\frac{I}{I_{\mathrm{ref}}}*\frac{A_{\mathrm{ref}}}{A}*\frac{n^{2}}{n_{\mathrm{ref}}^{2}}$$
where Φ_r_ is the quantum yield of the reference compound (quinine-sulfate in 0.1 mol/L perchloric acid, absolute quantum efficiency (Φ_r_ = 55%)), n is the refractive index of the solvent, I is the integrated fluorescence intensity, and A is the absorbance at the excitation wavelength. The absorbances at the wavelength of excitation were kept below A = 0.1 in order to avoid inner filter effects.

For UV–vis and fluorescence measurements, the investigated compounds were dissolved in acetonitrile at a concentration of 1.19 mM and were diluted to 2.38 × 10^−5^ M and 4.76 × 10^−6^ M in the solvents of interest. The spectra were processed using Spectragryph software [40].

### 3.1. Synthesis

Synthesis of N-(5-isocyanonaphthalen-1-yl)formamide (ICNF):

To the mixture of 1,5-diisocyanonaphtalene (described in the SI; 150 mg, 0.84 mmol, 1 equivalent), acetonitrile (3 mL) and distilled water (0.15 mL, 10 equivalent), trifluoroacetic acid (120 μL, 1.57 mmol, 1.9 equivalents) was added dropwise at 0 °C over 80 min. The ratio of the mono-(ICNF) and diformamide (NDF) products was monitored by HPLC-MS. The product ICNF was isolated by preparative HPLC (gradient elution, 15% → 45% acetonitrile in aqueous ammonium-hydrogen-carbonate solution (0.4 g in 1 L water)). ICNF was isolated as white powder (29 mg, yield: 17%, HPLC purity: >95%).
^1^H NMR (400 MHz, DMSO-*d*_6_) *δ* = 10.68 (s, 1H, H_9_), 10.51 (s, 1H, H_7_), 8.64 (d, *J* = 5.6 Hz, 1H, H_10_), 8.51 (d, *J* = 1.6 Hz, 1H, H_8_), 8.30 (dd, *J* = 8.8, 3.9 Hz, 2H, H_4_), 8.16 (d, *J* = 7.6 Hz, 1H, H_3_), 7.92 (dd, *J* = 12.9, 8.1 Hz, 4H, H_1_, H_6_), 7.80–7.69 (m, 2H, H_2_), 7.69–7.59 (m, 3H, H_5_, H_11_) ppm (Appendix A).^13^C NMR (101 MHz, DMSO-*d*_6_) *δ* = 160.5, 133.6, 128.5, 127.9, 125.7, 125.6, 125.4, 124.4, 120.9, 119.3, 118.8 ppm (Appendix A).HRMS (ESI/Q-TOF) *m*/*z*: [M+H]^+^ Calc. for C_12_H_9_N_2_O 197.0709; found 197.0652.

### 3.2. Density Functional Theory (DFT) Calculations

To obtain an explanation for the spectral changes which takes into account the electronic structure of the species, a previously tested calculation protocol was adopted. The geometry optimization of solvated molecules (S_0_) was carried out using M06-2X [41] density function combined with the classic 6-311++G(2d,2p) [42]. The solvent effect on the geometries was mimicked by integral equation formalism of the polarizable continuum model (IEF-PCM) [43] and the solvent cavity for seven solvents was constructed as implemented in Gaussian16 software package [44]. Normal mode analysis was performed to ensure that optimized geometry corresponded to real minima of the potential energy surface (noted as S_0_). The harmonic vibrational wavenumbers were used to obtain the thermochemical properties. The first singlet vertical excitation energies were computed for each molecule by time-dependent (TD) counterpart of TD-M06-2X/6-311++G(2d,2p), obtaining (λ_ex, max_). The maximum of the emission spectra (λ_em, max_) was approximated as the geometry optimization of the first singlet excited states (S_1_) of molecules at TD-M06-2X/6-311++G(2d,2p) level of theory. The electrostatic potentials of the studied molecules were also calculated by using the TD-M06-2X/6-311++G(2d,2p) level of theory.

## 4. Conclusions

It was established that, contrary to its simple and symmetric structure, 1,5-diisocyanonaphthalene possesses unique optical and chemical properties. The UV–vis absorption and emission properties are mostly unaffected by the medium. Its high energy absorption and emission (ν_abs, max_ = 300 nm and ν_em, max_ = 330, 345 nm) spectra makes it a promising candidate for the development of a turn-on probe for the detection of chemical species that can interact with or convert one of the isocyano groups. Using the controlled hydrolysis of DIN, we managed to prepare the nonsymmetric isocyano-formamido derivative N-(5-isocyanonaphthalen-1-yl)formamide (ICNF) for the first time. Starting from the respective di(tri, tetra, etc.)isocyanides, the acid-induced hydrolysis reaction may also be used to prepare other aromatic dyes bearing both isocyano and formamide groups in different substitution position, resulting in a new group of smart fluorescent probes. In contrast to DIN, the monoformamide (ICNF) turned to be sensitive to solvent polarity with a solvatochromic range of approximately 50 nm. Specific dye–solvent interactions, namely H-bonding, were discovered and proved by sophisticated quantum chemical (DFT) calculations. Titrations with pyridine further supported the presence of H-bonding. Moreover, the presence of ICNF dimers in hexane from the downward curvature of the Stern–Volmer plot was suggested and proved by high-level DFT calculations. Since formamide is the first intermedier in living organisms from isocyanides, results of these specific interactions may be valuable in further biological studies.

## Figures and Tables

**Figure 1 ijms-24-07780-f001:**
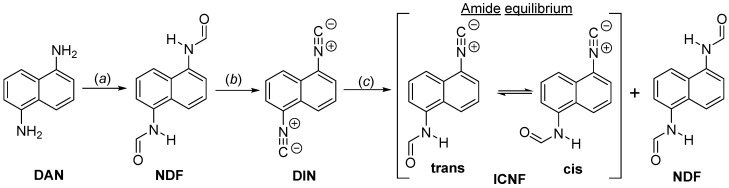
The synthesis scheme and name of the compounds used in this study. 1,5-diaminonaphthalene (DAN), 1,5-diisocyanonaphthalene (DIN), N-(5-isocyanonaphthalen-1-yl)formamide (ICNF) and N,N′-(naphthalene-1,5-diyl)diformamide (NDF). (**a**) Formic acid, toluene, 30 min, MW 120 °C; (**b**) POCl_3_, THF, TEA, 60 min, RT; (**c**) H_2_O, TFA, MeCN, RT. The amide cis-trans equilibrium is also shown.

**Figure 2 ijms-24-07780-f002:**
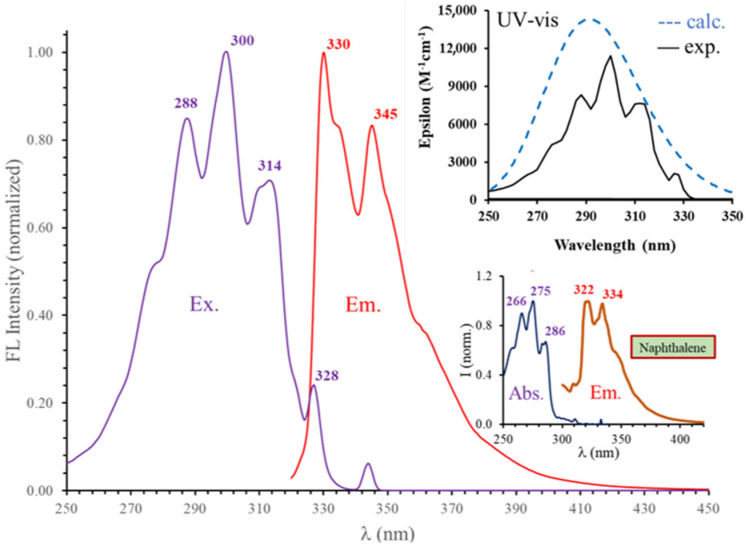
The normalized excitation (Ex.), emission (Em.), and UV–vis (inset, top) spectra of 1,5-diisocyanonaphthalene in acetonitrile ([DIN] = 1.197 × 10^−5^ M, T = 20 °C, λ_ex_ = 314 nm). The bottom inset shows the absorption and emission spectra of naphthalene recorded in cyclohexane [31].

**Figure 3 ijms-24-07780-f003:**
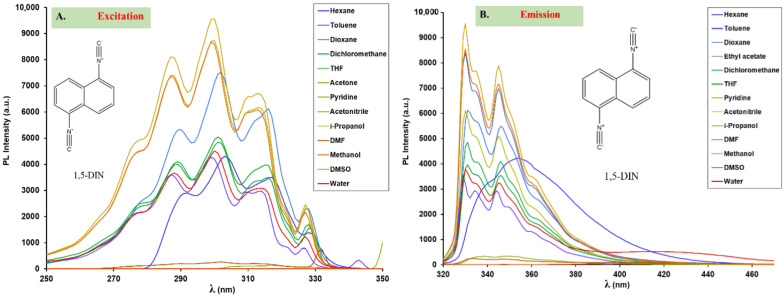
The excitation (**A**), emission (**B**) spectra of 1,5-diisocyanonaphthalene (DIN) in different solvents ([DIN] = 1.197 × 10^−5^ M, T = 20 °C).

**Figure 4 ijms-24-07780-f004:**
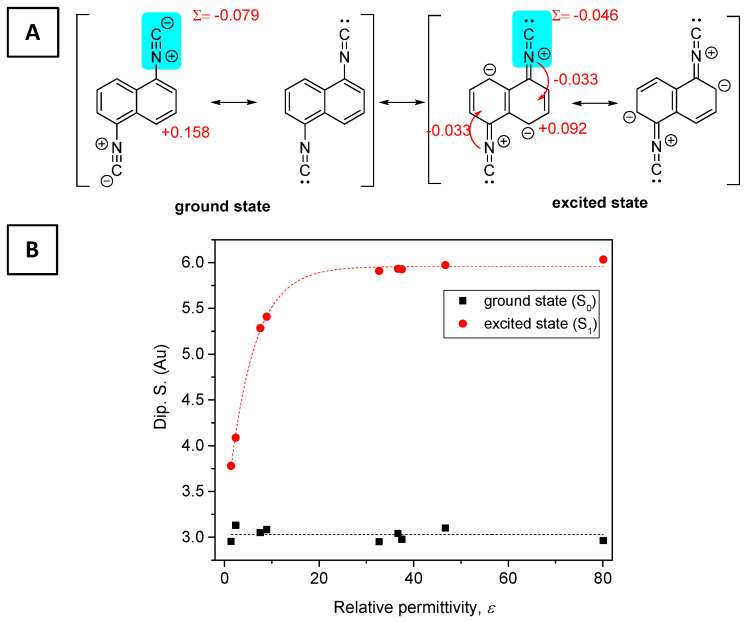
(**A**) The most representative resonance structures of DIN for the ground state (S_0_, left) and the excited state (S_1_, right). (**B**) The transition electric dipole moment (Dip.S.) at ground (S_0_, black) and excited states (S_1_, red) as a function of relative permittivity (ε) of the solvent applied.

**Figure 5 ijms-24-07780-f005:**
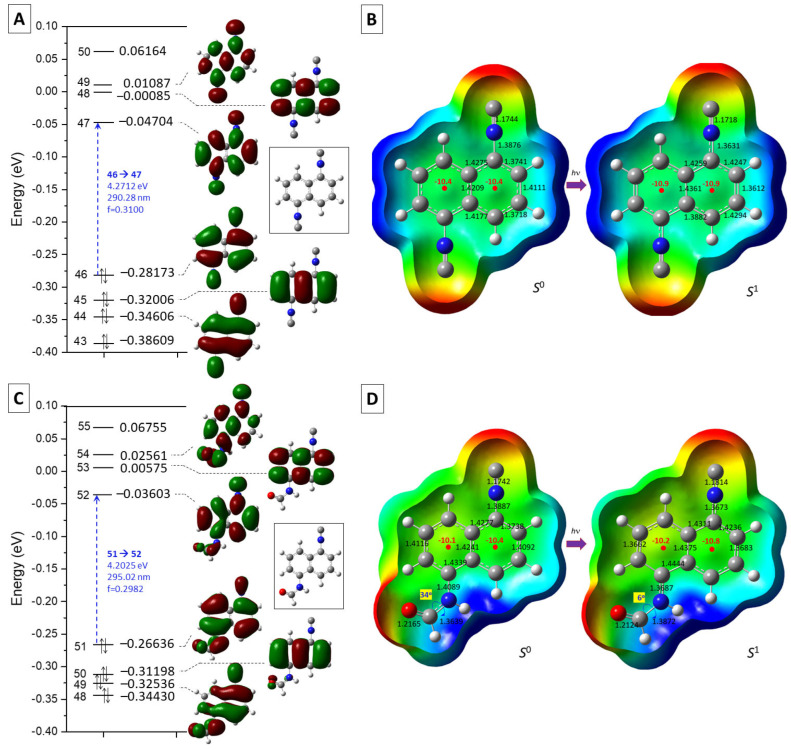
Representation of the energies and shapes of molecular orbitals (HOMO-3 LUMO-3) for DIN (**A**) and 1,5-ICNF (**C**) as well as the electrostatic potentials (ESP), of DIN (**B**) and 1,5-ICNF (**D**) mapped on the electron density surface. Red and blue colors mean the negatively and positively charged areas on ESP, respectively.

**Figure 6 ijms-24-07780-f006:**
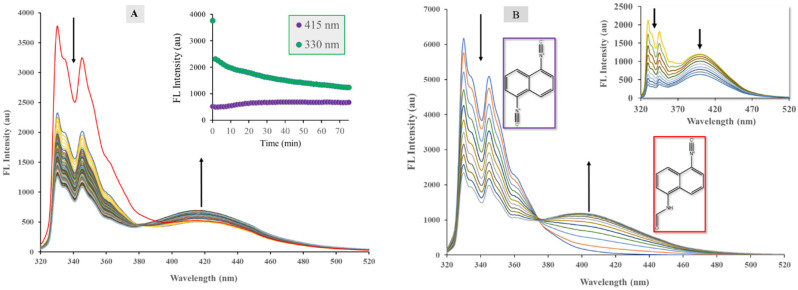
Spectral changes upon the reaction of 1,5-diisocyanonaphthalene (DIN) with 10% HNO_3_ in water (**A**) and with trifluoroacetic acid in acetonitrile (**B**). ([DIN] = 1.197 × 10^−5^ M, [HNO_3_] = 0.005 M, [TFA] = 0.10 M, T = 20 °C, λ_ex_ = 314 nm, Δt = 1.5 min (**A**), 3 min (**B**)). The inset graph (**B**) shows the final part of the reaction.

**Figure 7 ijms-24-07780-f007:**
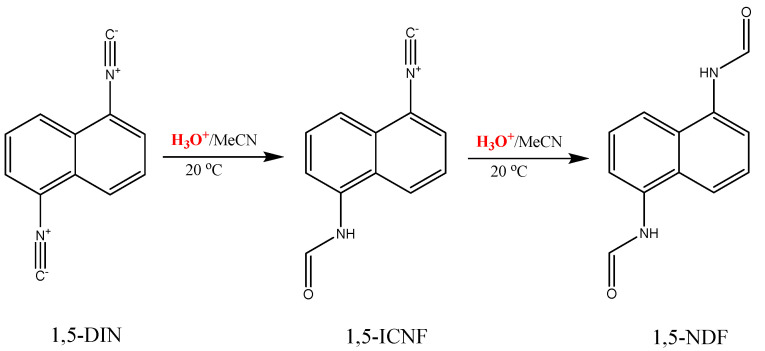
The hydrolysis reaction of 1,5-DIN, resulting in N-(5-isocyanonaphthalen-1-yl)formamide (1,5-ICNF) and N,N′-(naphthalene-1,5-diyl)diformamide (1,5-NDF).

**Figure 8 ijms-24-07780-f008:**
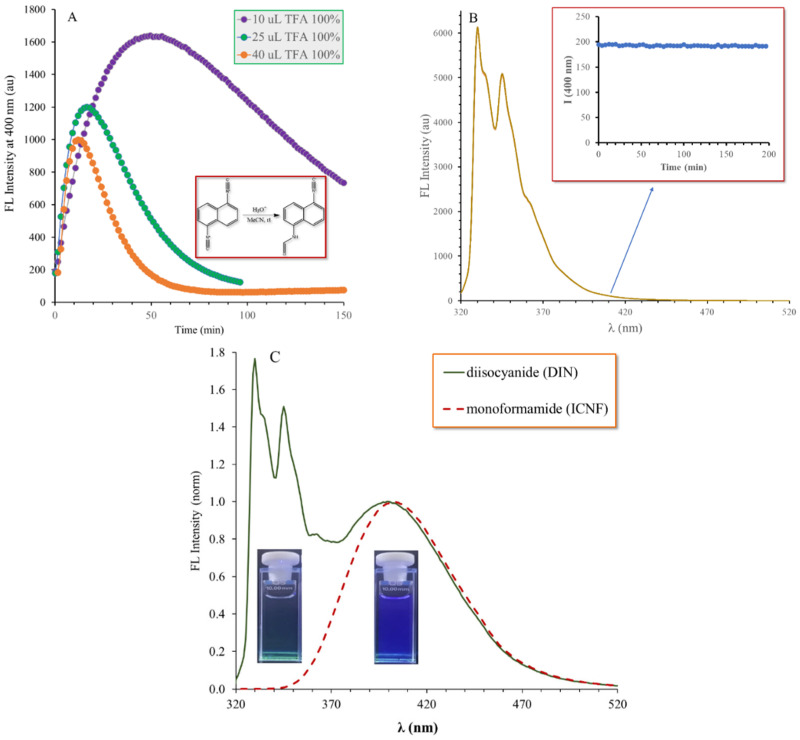
Changes in the emission intensity at 400 nm during the reaction of 1,5-diisocyanonaphthalene (DIN) with different amounts of trifluoroacetic acid (TFA) in acetonitrile (**A**) and without the addition of acid (**B**) ([DIN] = 1.197 × 10^−5^ M, T = 20 °C, λ_ex_ = 314 nm, Δt = 1.5 min). The (**C**) figure shows the normalized overlay emission spectra of DIN (solid, green) and the prepared 1-amino-5-formamido-naphthalene (red, dashed). The photos show the same solutions under UV irradiation (λ_ex_ = 254 nm).

**Figure 9 ijms-24-07780-f009:**
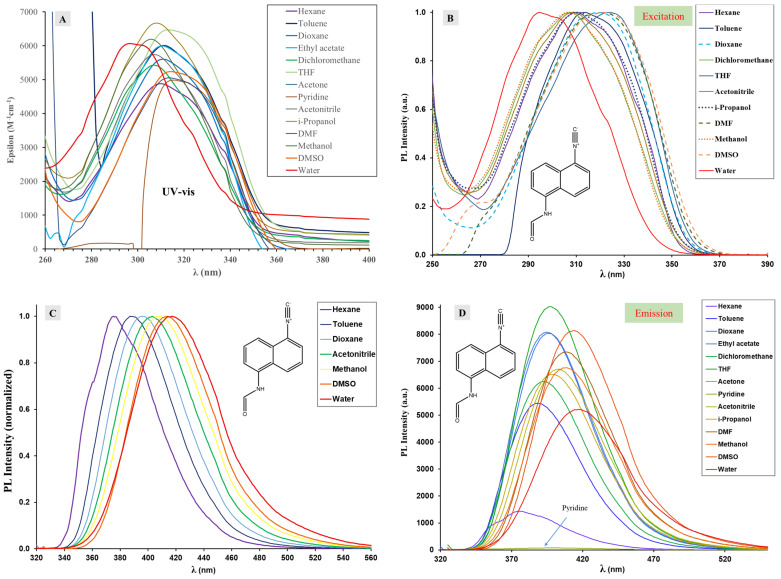
(**A**) UV–vis, (**B**) normalized excitation, (**C**) normalized emission, (**D**) emission spectra of N-(5-isocyanonaphthalen-1-yl)formamide (1,5-ICNF) recorded in solvents of different polarity ([ICNF] = 1.69 × 10^−5^ M, T = 20 °C).

**Figure 10 ijms-24-07780-f010:**
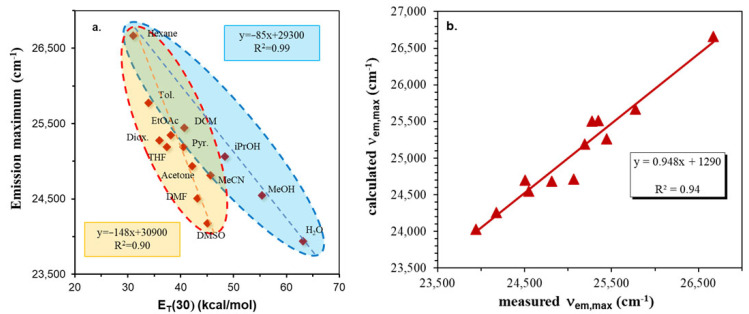
Variation of the fluorescence emission maximum with the empirical solvent polarity parameter E_T_(30) (**a**) and the Catalán (**b**) plots for the N-(5-isocyanonaphthalen-1-yl)formamide (1,5-ICNF) dye.

**Figure 11 ijms-24-07780-f011:**
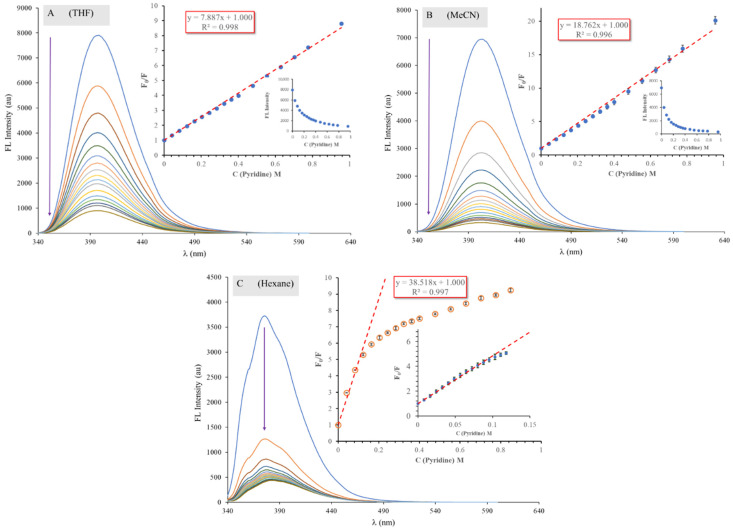
Spectral changes of ICNF upon titration with pyridine (10 µL or 1 µL for recording the initial part of the SV-curve in hexane) in various solvents: tetrahydrofuran (**A**), acetonitrile (**B**) and in hexane (**C**). The insets show the respective Stern–Volmer plots ([ICNF] =1.69 × 10^−5^ M, T = 20 °C, V_0_ = 3000 µL).

**Figure 12 ijms-24-07780-f012:**
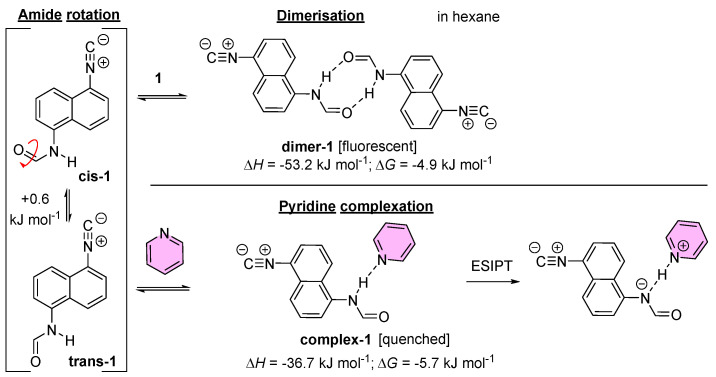
The thermodynamic representation of the equilibria in hexane solution. Left side shows the cis-trans amide rotamer equilibrium, where the trans amide is beneficial by only 0.6 kJ mol^−1^. The right top represents the exothermic dimerization of the ICNF, while the right bottom is the exothermic complexation by the added pyridine.

**Table 1 ijms-24-07780-t001:** The absorption maximum wavelengths (λ_Abs_), the molar extinction coefficients at λ_Abs_ (ε) the fluorescence emission maxima (λ_Em_), and the quantum yields (Φ_f_) determined in various solvents for 1,5-diisocyanonaphthalene (DIN). The dielectric constants of the solvents (ε_r_) are listed next to the solvent names.

Solvent (ε_r_)	Experimental				Computed	
	λ_Abs_ (nm)	ε × 10^−3^ (M^−1^cm^−1^)	λ_Em_ (nm)	Φ_f_ (%)	λ_Abs_ (nm)	λ_Em_ (nm)
n-Hexane (1.89)	300	10.3	329, 344	13.2	290	349
Dioxane (2.25)	302	14.2	331, 346	17.7	nc	nc
Toluene (2.38)	302	11.0	352	21.4	291	352
EtOAc (6.02)	300	14.9	330, 345	15.7	nc	nc
THF (7.58)	300	11.6	331, 346	14.8	291	360
DCM (8.93)	302	14.6	331, 346	10.0	291	361
Pyridine (12.4)	304	9.8	337, 352	0.7	nc	nc
2-Propanol (17.9)	300	13.3	330, 345	19.4	nc	nc
Methanol (32.7)	300	12.0	330, 345	18.3	290	365
DMF (36.7)	302	13.3	333, 347	1.1	291	365
Acetonitrile (37.5)	300	11.4	330, 345	20.6	290	365
DMSO (46.7)	303	11.3	-	0.5	291	365
Water (80.1)	300	4.0	330, 345	-	290	366

nc: not calculated.

**Table 2 ijms-24-07780-t002:** The absorption maximum wavelengths (λ_Abs_), the molar extinction coefficients at λ_Abs_ (ε), the fluorescence emission maxima (λ_Em_), and the quantum yields (Φ_f_) determined in various solvents for N-(5-isocyanonaphthalen-1-yl)formamide (1,5-ICNF). The dielectric constants of the solvents (ε_r_) are listed next to the solvent names (see Figure 9).

Solvent (ε_r_)	1,5-ICNF									
	λ_Abs_ (nm)	ε × 10^−3^ (M^−1^cm^−1^)	λ_Ex_ (nm)	λ_Em_ (nm)	Φ_f_	Δν_SS_ (cm^−1^)	λ_Abs_ (nm) Implicit Calc.	λ_Em_ (nm) Implicit Calc.	λ_Abs_ (nm) Explicit Calc.	λ_Em_ (nm) Explicit Calc.
n-Hexane (1.89)	310	4.9	310	375	0.11	5591	308	373	na	na
dimer in hexane									297	367
Dioxane (2.25)	310	5.6	318	396	0.55	6070	nc	nc	nc	nc
Toluene (2.38)	313	6.0	314	388	0.42	6176	305	375	nc	nc
EtOAc (6.02)	313	6.0	313	395	0.52	6600	nc	nc	nc	nc
THF (7.58)	314	6.5	324	397	0.55	6658	303	383	nc	nc
DCM (8.93)	308	5.4	307	393	0.52	7022	303	384	nc	nc
Pyridine (12.4)	316	4.9	316	397	0.005	6457	305	375	303	378
2-Propanol (17.9)	308	6.7	309	399	0.41	7405	na	nc	nc	nc
Methanol (32.7)	307	6.2	306	407	0.44	8081	302	387	295	388
DMF (36.7)	314	5.1	321	408	0.61	7343	302	387	301	392
Acetonitrile (37.5)	308	5.7	308	403	0.44	7591	302	387	nc	nc
DMSO (46.7)	314	5.2	322	414	0.72	7669	302	387	336	414
Water (80.1)	296	6.1	295	418	0.42	9843	302	388	292	385

nc: not calculated. na: not applicable.

**Table 3 ijms-24-07780-t003:** Solvent-independent correlation coefficients a_SA_, b_SB_, c_SP,_ and d_SdP_ of the Catalán parameters SA, SB, SP, and SdP, respectively, solute property of the reference system (emission maximum, ν_em, max_ and Stokes shift, Δν_ss_), correlation coefficient (R^2^), and number of solvents (n) calculated by multiregression analysis for the solvatochromism of ICNF.

	y_0_ (cm^−1^)	a_SA_	b_SB_	c_SP_	d_SdP_	R^2^	n
ν_em,max_	28,943	−1519	−982	−3729	−744	0.93	12
Δν_ss_	3416	3025	448	3722	561	0.93	12

## Data Availability

All data generated or analyzed during this study are included in this published article (and its Appendix A) or are available from the corresponding author on reasonable request.

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
