# Peer review of "Preparation and Optical Study of 1-Formamido-5-Isocyanonaphthalene, the Hydrolysis Product of the Potent Antifungal 1,5-Diisocyanonaphthalene"

_ijms, 2023, doi:10.3390/ijms24097780_

Round 1
Reviewer 1 Report
Review report
Article
Preparation and optical study of 1-formamido-5-isocy-anonaphthalene, the hydrolysis product of the potent anti-fungal 1,5-diisocyanonaphthalene
Manuscript ID: ijms-2340896
The above manuscript describes the importance and applications of 1-formamido-5-isocy-anonaphthalene in an attractive manner. This manuscript can be accepted with the following minor revisions.
1. Abstract should be presented in a brief manner. The first and second sentences in the abstract part were apt to mention in the introduction part. Generally 200 words were enough in this regard.
2. Author(s) have repeatedly using the terminology Asymmetric product for “1-formamido-5-isocyanonaphthalene (ICNF)” which was found to be an achiral aromatic compound. Is it unsymmetric or asymmetric?. Author(s) have to justify accordingly.
3. The influence of solvent in the excitation (A), emission (B) spectra of Figure 3 needs elaborated explanations. Similarly in Figure 9.
4. Are these products showing Antifungal activities? If possible include the results.
5. Generally Isocyanides can be obtained starting from amines in one step using a combination of KOH and CHCl3. But author(s) have chosen a two step process. Try to justify this point in the revised manuscript.
6. The following references are to be cited to increase the importance the present work.
(i) https://doi.org/10.1016/j.molliq.2021.117620
(ii) https://doi.org/10.21315/jps2020.31.3.3
(iii) https://doi.org/10.1002/slct.202203648
Author Response
Reviewer #1
Article
Preparation and optical study of 1-formamido-5-isocy-anonaphthalene, the hydrolysis product of the potent anti-fungal 1,5-diisocyanonaphthalene
Manuscript ID: ijms-2340896
The above manuscript describes the importance and applications of 1-formamido-5-isocy-anonaphthalene in an attractive manner. This manuscript can be accepted with the following minor revisions.
Thank You for Your positive comments! We appreciate it!
- Abstract should be presented in a brief manner. The first and second sentences in the abstract part were apt to mention in the introduction part. Generally 200 words were enough in this regard.
Thank You for this remark! You are right about the need for concise abstract part. However, a lot of people only read the abstract part during a literature search. We believe that the chemistry of aromatic isocyanoaminoarenes needs to be highlighted since it is a realtively new field and has a lot of potential in the field of sensor technology and pharmaceutical chemistry.
- Author(s) have repeatedly using the terminology Asymmetric product for “1-formamido-5-isocyanonaphthalene (ICNF)” which was found to be an achiral aromatic compound. Is it unsymmetric or asymmetric?. Author(s) have to justify accordingly.
Asymmetric was exchanged to nonsymmetric.
- The influence of solvent in the excitation (A), emission (B) spectra of Figure 3 needs elaborated explanations. Similarly in Figure 9.
Influence of the solvent effects are explained in the text as:
“As it is evident from Figure 2 and Figure 3A the UV-vis and the excitation spectra are identical for most solvents, indicating the absence of specific solvent-dye interactions (H-bonding or ESIPT) in the excited state. „
“Interestingly, fluorescence is completely quenched in DMSO ((Ff=0.5%), and very low fluorescence was observed in pyridine (Ff=0.7%) and DMF (Ff=1.1%), too (Figure 3A-B). This phenomenon is strange since all our previously prepared Isocyanaphtahelene (ICAN) derivatives were highly fluorescent in both DMSO and DMF [11,12]. Another strange behavior was observed in toluene, where instead of the double peak emission a broad charge transfer character band with a maximum of at lem,max=352 nm (Figure 3B) appeared. „
„The position of both the absorption and emission peaks remains virtually the same in every solvent (Table 1, Figure3A-B), since the dipole moment of DIN is 0 in both the S0 and S1 states owing to the symmetrical build of the molecule. The non-solvatochromic behavior and relatively high energy emission bands of DIN are advantageous for practical applications. „
- Are these products showing Antifungal activities? If possible include the results.
Naphthyl isocyanides show excellent in vivo antifungal activity as we demonstrated previously:
- Szigeti, Z. M.; Talas, L.; Szeles, A.; Hargitai, Z.; Nagy, Z. L.; Nagy, M.; Kiss, A.; Keki, S.; Szeman-Nagy, G., Potential Original Drug for Aspergillosis: In Vitro and In Vivo Effects of 1-N,N-Dimethylamino-5-Isocyanonaphthalene (DIMICAN) on Aspergillus fumigatus. J Fungi (Basel) 2022, 8, (10). DOI: 10.3390/jof8100985
- Nagy, M.; Szemán-Nagy, G.; Kiss, A.; Nagy, Z. L.; Tálas, L.; Rácz, D.; Majoros, L.; Tóth, Z.; Szigeti, Z. M.; Pócsi, I.; Kéki, S., Antifungal Activity of an Original Amino-Isocyanonaphthalene (ICAN) Compound Family: Promising Broad Spectrum Antifungals. Molecules 2020, 25, (4), 903. http://dx.doi.org/10.3390/molecules25040903
However, we have no information on the antifungal effects of the newly prepared Isocyanonaphthyl formamides, but we plan to investigate it soon.
- Generally Isocyanides can be obtained starting from amines in one step using a combination of KOH and CHCl3. But author(s) have chosen a two step process. Try to justify this point in the revised manuscript.
The introduction part was completed with the following sentence:
„However, using dichlorocarbene the yield of the diisocyano derivative is usually very low, therefore an alternative synthetic route, such as the formylation and subsequent dehydration of the starting diaminoarene molecule is needed, when the diisocyanoarene derivative is the product of interest.”
- The following references are to be cited to increase the importance the present work.
(i) https://doi.org/10.1016/j.molliq.2021.117620
(ii) https://doi.org/10.21315/jps2020.31.3.3
(iii) https://doi.org/10.1002/slct.202203648
The first two references have been included in the Introduction as:
“Moreover, isocyano groups are isoelectronic to cyano groups and could substitute them in specialty applications, such as liquid crystals [9,10].”
- Lakshmi Praveen, P., Estimation of absorption spectral shifts of cyano biphenyl liquid crystals: An impact of different solvents and oxygen substitution. Journal of Molecular Liquids 2021, https://doi.org/10.1016/j.molliq.2021.117620
- Nayak, S. K.; Praveen, P. L., Mesophase behaviour of a cyanobiphenyl molecule in polar aprotic solvent: Rigidity effect. Journal of Physical Science 2020, 31, (3), 33-45. https://doi.org/10.21315/jps2020.31.3.3
Reviewer 2 Report
The MS titled "Preparation and optical study of 1-formamido-5-isocy-anonaphthalene, the hydrolysis product of the potent anti-fungal 1,5-diisocyanonaphthalene" authored by Dr. Mucsi, Dr. Nagy and other co-workers, describes the controlled hydrolysis of 1,5-diisocyanonaphthalene (DIN) to synthesize asymmetric isocyano-formamido derivative N-(5-isocyanonaph-thalen-1-yl)formamide (ICNF). The authors studied UV-vis absorption and emission properties of this molecule in various mediums. The authors also performed quantum chemical (DFT) calculations along with titrations with pyridine to support the presence of H-bonding in this molecule. This manuscript is well written, the science is sound, and the approach is solid. The authors perform a deep and complete study to support this article. However, there are some parts of this paper that need improvement before acceptance, and I have the following questions and suggestions:
Comments:
1. 1H NMR spectra of ICNF (compound 4) exhibit twenty-six protons, however the molecule has eight protons in its molecular structure. Nevertheless, 13C spectra shows eleven peaks only. Detail assignment of the peaks along with a clear explanation is required about the extra peaks.
2. Page 2, line 2, the paragraph concludes as “most useful building-blocks of “smart-materials”. However, no citation is mentioned for this statement. Authors should insert proper citation for this statement.
3. On page 3, line 17, read as “very good agreement with the theoretical calculations (Fig1 inset)”. Authors should substitute “Fig 1 inset” to “Fig 2 inset”.
4. Page 8, line 8, read as “The results are presented in Figure 4A”. Authors should correct “Figure 4A” to Figure 6A.
5. The authors should change "amid rotation" to " amide rotation" in Figure 12.
6. In the reference section abbreviation for all the journals needed according to the Int. J. Mol. Sci. referencing style (e.g. ref. 23 European Journal of Organic Chemistry it should be Eur. J. Org. Chem.). The publication year for all the cited articles should be in bold letters (e.g. ref. 8, 10, 29), volume number will be in the italics (e.g. ref. 10). All the references need to be revised carefully.
Author Response
Reviewer #2
The MS titled "Preparation and optical study of 1-formamido-5-isocy-anonaphthalene, the hydrolysis product of the potent anti-fungal 1,5-diisocyanonaphthalene" authored by Dr. Mucsi, Dr. Nagy and other co-workers, describes the controlled hydrolysis of 1,5-diisocyanonaphthalene (DIN) to synthesize asymmetric isocyano-formamido derivative N-(5-isocyanonaph-thalen-1-yl)formamide (ICNF). The authors studied UV-vis absorption and emission properties of this molecule in various mediums. The authors also performed quantum chemical (DFT) calculations along with titrations with pyridine to support the presence of H-bonding in this molecule. This manuscript is well written, the science is sound, and the approach is solid. The authors perform a deep and complete study to support this article. However, there are some parts of this paper that need improvement before acceptance, and I have the following questions and suggestions:
Thank You for Your positive comments! We appreciate it!
Comments:
- 1H NMR spectra of ICNF (compound 4) exhibit twenty-six protons, however the molecule has eight protons in its molecular structure. Nevertheless, 13C spectra shows eleven peaks only. Detail assignment of the peaks along with a clear explanation is required about the extra peaks.
The 1H NMR spectra in the supporting information were exchanged to better assigned ones. In addition a COSY spectrum of ICNF was added as Figure S7.
The text has been completed with the following explanation:
“In addition, theoretical calculations could explain the unusual splitting of the signals in the 1H NMR spectrum of ICNF (Figure S5). Since the rotation around the C-N bond in amides is usually hindered, the appearance of two isomers (rotamers), namely cis and trans is expected (Figure 1 and 12). Based on DFT studies the energy difference between the two forms is 0.6 0.6 kJ×mol-1 the cis rotamer having lower energy. The 0.6 kJ×mol-1 difference is agrees well with the experimental cis:trans ratio of 70:30 determined from the signal ratios in the 1H NMR spectrum of ICNF.”
- Page 2, line 2, the paragraph concludes as “most useful building-blocks of “smart-materials”. However, no citation is mentioned for this statement. Authors should insert proper citation for this statement.
The sentence has been completed as:
„This unique behavior makes isonitrile one of the most useful building-block of “smart materials”, such as photochromic complexes [6], supramolecular networks [7] and environment sensitive dyes [8].”
- Lu, T.; Yang, X.; Wang, X.-Y.; Li, Z.; Yin, J.; Liu, S. H., Dithienylethene-bridged gold(I) isocyanide complexes: Syn-thesis, photochromism and “turn-on” fluorescent switching behavior. Dyes and Pigments 2021, 185. https://doi.org/10.1016/j.dyepig.2020.10893
- Mikherdov, A. S.; Popov, R. A.; Smirnov, A. S.; Eliseeva, A. A.; Novikov, A. S.; Boyarskiy, V. P.; Gomila, R. M.; Frontera, A.; Kukushkin, V. Y.; Bokach, N. A., Isocyanide and Cyanide Entities Form Isostructural Halogen Bond-Based Supramolecular Networks Featuring Five-Center Tetrafurcated Halogen···C/N Bonding. Crystal Growth & Design 2022, 22, (10), 6079-6087. https://doi.org/10.1021/acs.cgd.2c00686
- Zhou, N.; Zhang, Y.; Chen, Z.; Zhu, X.; He, X., Developing luminescent ratiometric thermometers based on co-polymers containing Platinum(II) isocyanide complex. Dyes and Pigments 2021, 184. https://doi.org/10.1016/j.dyepig.2020.108815
- On page 3, line 17, read as “very good agreement with the theoretical calculations (Fig1 inset)”. Authors should substitute “Fig 1 inset” to “Fig 2 inset”.
The numbering was corrected.
- Page 8, line 8, read as “The results are presented in Figure 4A”. Authors should correct “Figure 4A” to Figure 6A.
The Figure number was corrected.
- The authors should change "amid rotation" to " amide rotation" in Figure 12.
The typo was corrected.
- In the reference section abbreviation for all the journals needed according to the Int. J. Mol. Sci. referencing style (e.g. ref. 23 European Journal of Organic Chemistry it should be Eur. J. Org. Chem.). The publication year for all the cited articles should be in bold letters (e.g. ref. 8, 10, 29), volume number will be in the italics (e.g. ref. 10). All the references need to be revised carefully.
The reference formats were checked and corrected according to the comment of the reviewer.
Reviewer 3 Report
It is an interesting paper, but I am missing information regarding the characterization of some of the compounds: In the 1H-NMR data reported for the formamides, the number of protons are almost double the actual numbers of protons. The splitting of the signals is also not, what you would expect from a 1,5-disubstituted naphtalene; the 1 and 5 protons should be a higher ppm-value than the 2,3,4 and 6 protons. It is likely due to the restricted rotation around the amide-bond in the formamide, but it would be nice with COSY-spectra and a clarification on how the coupling patterns make sense. I.e. the ratios between the rotamers.
Author Response
Reviewer #3
It is an interesting paper, but I am missing information regarding the characterization of some of the compounds: In the 1H-NMR data reported for the formamides, the number of protons are almost double the actual numbers of protons. The splitting of the signals is also not, what you would expect from a 1,5-disubstituted naphtalene; the 1 and 5 protons should be a higher ppm-value than the 2,3,4 and 6 protons. It is likely due to the restricted rotation around the amide-bond in the formamide, but it would be nice with COSY-spectra and a clarification on how the coupling patterns make sense. I.e. the ratios between the rotamers.
Thank You for Your positive comments! We appreciate it!
The 1H NMR spectra in the supporting information were exchanged to better assigned ones. In addition, a COSY spectrum of ICNF was added as Figure S7.
The text has been completed with the following explanation:
“In addition, theoretical calculations could explain the unusual splitting of the signals in the 1H NMR spectrum of ICNF (Figure S5). Since the rotation around the C-N bond in amides is usually hindered, the appearance of two isomers (rotamers), namely cis and trans is expected (Figure 1 and 12). Based on DFT studies the energy difference between the two forms is 0.6 0.6 kJ×mol-1 the cis rotamer having lower energy. The 0.6 kJ×mol-1 difference is agrees well with the experimental cis:trans ratio of 70:30 determined from the signal ratios in the 1H NMR spectrum of ICNF.”
Round 2
Reviewer 3 Report
The manuscript has been revised according to the previous comments and is acceptable for publication.